# Study of *Helicobacter pylori* Isolated from a High-Gastric-Cancer-Risk Population: Unveiling the Comprehensive Analysis of Virulence-Associated Genes including Secretion Systems, and Genome-Wide Association Study

**DOI:** 10.3390/cancers15184528

**Published:** 2023-09-12

**Authors:** Batsaikhan Saruuljavkhlan, Ricky Indra Alfaray, Khasag Oyuntsetseg, Boldbaatar Gantuya, Ayush Khangai, Namsrai Renchinsengee, Takashi Matsumoto, Junko Akada, Dashdorj Azzaya, Duger Davaadorj, Yoshio Yamaoka

**Affiliations:** 1Department of Environmental and Preventive Medicine, Oita University Faculty of Medicine, Yufu 879-5593, Oita, Japan; saruuljavkhlan@yahoo.com (B.S.); rickyindraalfaray@gmail.com (R.I.A.); m20d9103@oita-u.ac.jp (A.K.); m22d9105@oita-u.ac.jp (N.R.); tmatsumoto9@oita-u.ac.jp (T.M.); akadajk@oita-u.ac.jp (J.A.); 2Helicobacter pylori and Microbiota Study Group, Institute of Tropical Disease, Universitas Airlangga, Surabaya 60286, East Java, Indonesia; 3Endoscopy Center, Mongolia Japan Hospital, Mongolian National University of Medical Sciences, Ulaanbaatar 14210, Mongolia; oyuntsetseg.kh@mnums.edu.mn (K.O.); gantuya@mnums.edu.mn (B.G.); 4Department of Gastroenterology and Hepatology, Mongolian National University of Medical Sciences, Ulaanbaatar 14210, Mongolia; azzaya2000@gmail.com (D.A.); davaadorj@mnums.edu.mn (D.D.); 5The Research Center for GLOBAL and LOCAL Infectious Diseases (RCGLID), Oita University, Yufu 870-1192, Oita, Japan; 6Department of Medicine, Gastroenterology and Hepatology Section, Baylor College of Medicine, Houston, TX 77030, USA

**Keywords:** gastric diseases, virulence factor, *cag* pathogenicity island, integrative and conjugative element (ICE), outer membrane proteins, population genomics, secreted proteins, Mongolia

## Abstract

**Simple Summary:**

*Helicobacter pylori* infection is known to be a major risk factor for gastric cancer, which continues to be a huge health burden worldwide, especially in Mongolia. There is a complicated interaction between host genetics, environmental factors, and bacterial virulence determinants in the etiology of this cancer. The virulence factors produced by this bacterium are among the most important contributors to the onset of gastric cancer. This study provides an accessible introduction to the relationship between *H. pylori* infection and gastric cancer and focuses on elucidating the impact of *H. pylori’s* varying virulence factors on a set of virulence-associated genes at the level of gene presence or absence and single nucleotide polymorphisms from a genome-wide *H. pylori* perspective on gastric disease progression. Current results provide the basis for gastric progression depending on bacterial factors and help to find new insights into the prevention and treatment of gastric cancer.

**Abstract:**

Background: The prevalence of gastric cancer in Mongolia, in East Asia, remains the highest in the world. However, most *Helicobacter pylori* strains in Mongolia have a less virulent Western-type CagA. We aimed to determine how *H. pylori* genomic variation affected gastric diseases, especially gastric cancer, based on comprehensive genome analysis. Methods: We identified a set of 274 virulence-associated genes in *H. pylori*, including virulence factor and outer membrane protein (OMP) genes, the type four secretion system gene cluster, and 13 well-known virulence gene genotypes in 223 *H. pylori* strains and their associations with gastric cancer and other gastric diseases. We conducted a genome-wide association study on 158 *H. pylori* strains (15 gastric cancer and 143 non-gastric cancer strains). Results: Out of 274 genes, we found 13 genes were variable depending on disease outcome, especially iron regulating OMP genes. *H. pylori* strains from Mongolia were divided into two main subgroups: subgroup (Sg1) with high risk and Sg2 with low risk for gastric cancer. The general characteristics of Sg1 strains are that they possess more virulence genotype genes. We found nine non-synonymous single nucleotide polymorphisms in seven genes that are linked with gastric cancer strains. Conclusions: Highly virulent *H. pylori* strains may adapt through host-influenced genomic variations, potentially impacting gastric carcinogenesis.

## 1. Introduction

*Helicobacter pylori* is a urease-positive, Gram-negative bacterium that infects approximately four billion people and is the causal agent for severe diseases, from peptic ulcer disease (PUD) to gastric cancer (GC) [1,2,3,4]. Studies suggest that, besides host factors (e.g., genetics) and environmental factors (e.g., lifestyles, and diets such as high salt and alcohol consumption), *H. pylori* infection becomes a leading risk factor for GC development [5]. One of the important key players, the virulence factors of *H. pylori* may contribute to GC development, especially in high-risk regions where the host and environmental factors support the development of cancer [6,7]. Notably, the rate of *H. pylori* infection and the prevalence of GC vary ethnographically [8,9]. For instance, it was believed that Western countries harbored less virulent strains of *H. pylori*, leading to a lower occurrence of GC compared to East Asian countries [10,11]. It has been hypothesized that the reason for the high GC incidence in East Asia is that this region has various potent virulence factor genotypes, such as CagA, VacA, and OipA [4,12,13]. Mongolia, located in East Asia, has the highest burden of GC in the world, and the seroprevalence of *H. pylori* infection rate was 80.0% in 2018 [7]. Interestingly, most of the Mongolian dyspeptic patients are infected with less-virulent strains, like a Western-type CagA, and many cases of GC were found in the upper part of the stomach [12,14], although most GC arises in the middle or lower part of the stomach in other East Asian countries infected dominantly by *H. pylori* strains with East Asian-type CagA, such as Japan, Korea, and China [14]. This paradoxical phenomenon can be termed the “Mongolian enigma”. Thus, it is necessary to conduct an in-depth study to find the reason for the highly virulent *H. pylori* despite possessing Western-type CagA in Mongolia.

The *H. pylori* genome generally has 1.65 million base pairs in length and encodes around 1600 proteins [15]. The genome plasticity in *H. pylori* is relatively high, and numerous genes can be turned “on” and “off” by repeated nucleotide mutagenesis [15]. Many important genes related to this bacterial virulence are currently being studied. These virulence-associated genes include a variety of genes related to secreted proteins and non-secreted proteins, such as outer membrane proteins (OMPs), adhesins, secreted enzymes, secretion systems, and effector proteins [16]. Moreover, many secreted proteins interact with host factors and immune systems, thereby enhancing the virulence and efficacy of the infection in the host environment [16]. Unfortunately, to date, the majority of their effects and relationships with GC and other diseases are still largely unknown. Understanding *H. pylori*’s virulence-associated genes roles at different stages of gastric disease might help find successful treatments. Especially, eradication by identifying more virulent strains may prevent GC development in countries where the cancer risk and *H. pylori* prevalence are high.

Correlations between *H. pylori* ancestry and host disease phenotypes are another pressing concern. Seven unique *H. pylori* populations (“hp”) have been discovered using multi-locus sequencing typing (MLST) and Structure analysis, and these populations are named after the continents in which they were first found: hpAfrica2, hpAfrica1, hpNEAfrica, hpEurope, hpAsia2, hpEastAsia, and hpSahul [17]. More populations and subpopulations (“hsp”) were identified by subsequent FineStructure whole genome-based analysis. For instance, hpNorthAsia contains hspIndigenousAmericas and hspAltai, which are more often seen infecting indigenous populations in some Asian and American countries [18]. The risk of developing GC has been shown in several studies to track the prevalence of *H. pylori* populations [19,20]. In contrast, most hspEAsia strains of hpEastAsia have been infected in East Asian countries, which also have the highest rates of GC [21]. However, the majority of these studies have only used MLST analysis, which only employs seven housekeeping genes, to compare host disease phenotypes and *H. pylori* ancestry.

Some studies attempted to understand the relationship between *H. pylori* and GC development by performing genome-wide association studies (GWAS). It is well known that the results of GWAS can be similar or different depending on the bacterial population. For instance, two different GWAS studies conducted in hpEurope [22] and hspEAsia [23] populations determined 32 and 11 significant single nucleotide polymorphisms (SNPs), respectively. Notably, there were no matched hits between these two studies due to population differences and comparison groups. However, these results on other genotypes suggest that variation related to this disease differs between bacterial populations. The GWAS on *H. pylori* in other populations has never been studied.

To address these queries, we analyzed the 223 genomes of *H. pylori* clinical isolates and the demographic and clinical data of patients to identify genomic variability and mutations related to GC and other gastric diseases. Therefore, we comprehensively analyzed the set of virulence-associated genes, including the type IV secretion system (T4SS), at gene presence and their genotyping level in patients with GC and other diseases for the identification of disease-associated genetic variants of the *H. pylori* subgroup, which infected the Mongolian population at high risk of developing GC. Furthermore, we conducted a GWAS analysis to discover new candidates for novel genetic variations and oncoproteins related to the oncogenic process. This study highlights the importance and correlation among the virulence-associated genes, T4SS, and other genes known to be associated with GC and other diseases in the high-gastric-cancer-risk population.

## 2. Materials and Methods

### 2.1. Study Population and Sampling

A total of 223 *H. pylori* strains (*H. pylori*-Mon) from 207 Mongolian patients with non-GC and 16 patients with GC were successfully cultured from gastric biopsy specimens of these patients who resided in five different regions of Mongolia: Ulaanbaatar, the capital city, the Western part (Uvs province), the Northern part (Khuvsgul province), the Southern part (Umnugovi province), and the Eastern part (Khentii province) between November 2014 and August 2016. Out of 223 *H. pylori* strains, 88 strains were used from our previous study and 135 strains were newly sequenced for this study. During esophagogastroduodenoscopy, four biopsy specimens were taken for *H. pylori* culture and histological examination (antrum: approximately 2 cm from the pyloric ring in the greater curvature, angulus: lesser curvature, and corpus: greater curvature, 8–10 cm from the esophagogastric junction) [24].

### 2.2. H. pylori Isolation and Histological Evaluation

All samples were shipped immediately to the Department of Environmental and Preventive Medicine, Oita University, for *H. pylori* detection, culture, sequencing, and histological evaluation. All biopsy samples were determined to be positive for *H. pylori* infection and assessed for mucosal damage by morphological analysis and immunohistochemistry, as previously described [12]. We followed the updated Sydney classification system for gastritis, which divided non-malignant mucosal status into four grades: 0 for “normal”, 1 for “mild”, 2 for “moderate”, and 3 for “marked” for degrees of inflammation by neutrophil and monocyte infiltrations in the lamina propria or superficial epithelium, atrophy by glandular loss, and metaplasia by intestinal metaplasia of mucosal epithelium. Positive samples were graded 1 or above [25].

After homogenization, the biopsy samples were plated on commercial selective *H. pylori* plate (Nissue Pharmaceutical Co., Ltd., Tokyo, Japan), followed by incubation for up to 10 days at 37 °C under microaerophilic conditions (5% O_2_, 10% CO_2_, and 85% N_2_). *H. pylori* colonies are selected after morphology identification, and genomic DNA extraction was done using the DNeasy Blood and Tissue Kit (Qiagen, Valencia, CA, USA) following the manufacturer’s instructions.

### 2.3. Genome Sequencing, De Novo Assembly, and Annotation

A genomic DNA sequencing of 135 new *H. pylori* strains was performed using the Miseq platform (Illumina, Inc., San Diego, CA, USA), allowing a paired-end (2 × 300 bp) sequencing strategy, and library preparation was done with the TruSeq Nano DNA High Throughput Library Prep Kit (Illumina, Inc., San Diego, CA, USA). After quality checking the raw reads using FastQC version 0.11.9 (https://github.com/s-andrews/FastQC, accessed on 23 August 2022), Trimmomatic pipeline version 0.39 was used to remove low-quality reads, resulting in averages of Q20. Then, we performed de novo assembly using Spades assembler implanted with Shovill version 1.1.0 (https://github.com/tseemann/shovill, accessed on 23 August 2022) and did quality assurance on all 223 draft genomes using the Quality Assessment Tool for Genome Assemblies (QUAST) version 5.2.0 [26]. All draft genomes were annotated for contigs > 500 bp with Prokka version 1.14.659 to identify features of interest in genomic DNA sequences [27].

### 2.4. Identification of Virulence-Associated Genes and T4SS Genes

A wide range of *H. pylori* virulence-associated genes (HpVAG, which are listed in Appendix A), was made so that all reported secreting proteins and virulence factor genes variables for *H. pylori* species could be looked at in depth. All genes of HpVAG were sourced from the VFDB database via Abricate pipeline version 1.0.1 and previously published research [28,29,30]. A total of 274 genes were created and organized into functional groups such as OMPs (*n* = 64), and virulence factors (*n* = 210). The virulence factor genes contain flagellar components (*n* = 62), binding or transport proteins (*n* = 11), outer membrane biogenesis complex components (n = 26), VacA and other toxins (*n* = 6), immunomodulation proteins (*n* = 10), stress survival (*n* = 10), redox systems and electron transport chain enzymes (*n* = 17), putative solenoid protein enzymes (*n* = 9), proteases (*n* = 7), other enzymes (*n* = 25), hypothetical proteins (*n* = 27). These hypothetical protein genes have been experimentally demonstrated to be secreted proteins previously [28]. The distributions of each gene of HpVAG in all *H. pylori*-Mon were determined by calculating the percentage of all *H. pylori*-Mon numbers and each genotype distribution by calculating the proportion of all presented gene numbers.

To identify the Integrative and Conjugative components of *Helicobacter pylori type 4 system* (ICE*Hptfs*) type of T4SS and completeness in *H. pylori*-Mon, we employed a *vir* homolog gene cluster of *tfs* from previously used reference strains representing ICE*Hptf* [30], such as the Gambia94/24, India7 (*tfs3*), P12 (ICE *tfs4a*), strain Shi470 (*tfs4b*), SAfrica7 (*tfs4c*), and 26,695 (*cag*PAI and *comB*) strains.

In order to characterize the HpVAG and T4SS gene profiles, we performed a BLASTn based Abricate pipeline to ascertain whether or not *H. pylori*-Mon had virulence genes. All draft genomes were blasted against the corresponding genes of HpVAG, and the threshold is set at an e-value of 1 × 10^−20^, 70% sequence similarity, and 50% coverage with the query sequence.

Due to *H. pylori* OMP families having high similarity intra-family genes (Appendix A), sequence similarity of 80% and query coverage of 50% were utilized as the thresholds for the determination of all OMP genes.

### 2.5. Characterization and Genotyping of Well-Known Virulence-Associated Genes

To describe the genotype of well-known virulence-associated genes, we characterized the genotypes and on/off status of *cagA*, *vacA*, *oipA*, *babA*, *babB*, *babC*, *hopZ*, *hopQ*, *sabA*, *sabB*, *iceA*, *homA/B*, and *htrA.* Briefly, initially, all sequences of *H. pylori*-Mon performed reference mapping using BLASTn [31] against reference genes of HpVAG to find a single-best hit search and manually curated genotypes of each gene based on previously reported nucleotide or amino acid identity using CLC Genomics Workbench version 20.0.4 software. Further investigation delved into the examination of the variations within the cytosine-thymidine (CT) di-nucleotide repeat motifs positioned at the signal sequences of the *babB*, *hopZ*, *oipA*, *sabA*, and *sabB* genes. The point of this investigation was to evaluate the functional status of the genes identified in previous publications, as supported by experimental evidence [32,33,34,35]. The genes in the “on” state undergo translation to produce fully functioning proteins, whereas the genes in the “off” state result in the production of immature proteins.

### 2.6. Population Structure and GWAS

To define the *H. pylori* population at the fine-scale population structure, we used previously described *H. pylori* 65 reference strains (Appendix A) for the FineStrucuture analysis. Initially, the core SNP sites were called against *H. pylori* reference strain 26,695 (NC000915.1) by the Snippy 4.2.7 pipeline, and 456,004 possible core SNP sites were analyzed from a total of 288 strains (*H. pylori*-Mon and references strains). Then, the Chromopainter algorithm reconstructed individual haplotypes by “paintings” and the resulting co-ancestry matrix clustering by employing a Bayesian approach for 200,000 iterations of both burn-in and Markov chain Monte Carlo (MCMC). To visualize the results, we made a dendrogram using the R script, which was implemented with the Finestructure pipeline.

The GWAS was conducted by a pipeline based on the Pyseer package (version 1.3.11), following a previously reported method. In short, we used two approaches, such as SNP-based and k-mer or unitig-based (short motifs of a fixed length) approaches to detect GC specific genomic variations of *H. pylori*. The significance of these two approaches was calculated by a linear mixed model. We perform GWAS using core SNPs called by the Snippy pipeline, and k-mers are generated using the unitig-caller pipeline (version 1.3.0) [36]. Significant SNPs and k-mers were annotated using thresholds of *p* value < 10^−6^ and *p* value < 10^−8^ with allele frequencies different at 20%, and an allele frequency greater than 50% or less than 5% was excluded since they represented the predominant or rare alleles. We mapped all significant SNPs against the reference strain 26,695 genome for visualization with a Manhattan plot.

Non-synonymous SNP-mapped genes underwent the construction of de novo three-dimensional (3D) structural protein models employing the ColabFold v1.5.2-patch:AlphaFold2 with MMseqs2 [37], referencing the proteins of the 26,695 strain. Visualization of these models was achieved using the PyMol Molecular Graphics System, version 2.5.2 (Schrödinger LLC, New York, NY, USA).

### 2.7. Statistical Analysis

The R Studio software version 4.0.2 was used to analyze the data. Depending on the data type and number of comparisons, differences between groups were compared using Chi-Square tests, Fisher’s exact test, Mann–Whitney test, or the Kruskal–Wallis’s test. To evaluate the risk score, we used logistic regression analysis. The *p* value of 0.05 was used to determine statistical significance. All *p* values were two sided.

## 3. Results

### 3.1. Characteristics of Clinical Isolates H. pylori, and Their Population Structure

We summarize the general characteristics of 223 patients’ demographic data in Appendix A. The patients’ average age was 45.2 years, and 70.9% were female. Based on histological assessment, the patients can be divided into four disease groups: 120 (53.8%) non-atrophic gastritis (NAG, Sydney grade 0–1), 9 (4.0%) peptic ulcer disease (PUD), 78 (35.0%) atrophic gastritis (Sydney grade > 1) or intestinal metaplasia (AG/IM), and 16 (7.2%) gastric cancer (GC).

The substantial variations and divergence in genome across *H. pylori* phylogeographic populations are notable and undeniable [17]. Therefore, we used FineStructure analysis to describe *H. pylori*-Mon strains so that we could better comprehend the pathogenic participation of virulence-associated genes. After comparing with previously deposited *H. pylori* comprising 65 reference strains from 14 different subpopulations relative to our *H. pylori*-Mon strains, interestingly, there were two main subgroups identified, such as 158/223 (70.9%) subgroup 1 (Sg1) and 45 (20.2%) Sg2 (hspAltai subpopulation) strains. Additionally, 15 (6.7%) of the strains (Sg3) were clustered with hspNEurope, 4 (1.8%) with hspEAsia, and 1 (0.4%) with hpAsia2 in Figure 1.

Surprisingly, nearly all *H. pylori* strains 15/16 (93.8%) from GC cases were assigned to Sg1 strains (Table 1). Sg1 strains were a highly virulent subgroup (OR = 6.7, *p* = 0.044) to GC among *H. pylori*-Mon, while hspAltai was a less virulent subgroup and the majority group (20.2%) among the remaining *H. pylori* subgroups. These two subgroups were selected for further analysis in order to facilitate a comparison of the set of HpVAG and genotypes.

### 3.2. Diversity of OMP and Virulence Factor Genes Associated with Gastric Cancer and Other Diseases

*H. pylori*’s pathogenic processes are intricate and variable in response to changes in the expression of genomic variation and virulence genotypes of genes. We first looked at how effectively the presence or absence of genes conveyed diversity in certain gastric diseases. Figure 2 shows the status of 210 and 64 (with four duplicated homologous OMPs) genes in two main *H. pylori* subgroups in *H. pylori*-Mon strains and four diseases (CG, PUD, AG/IM, GC) compared to HpVAG, which included genes of OMP and virulence factor separately.

We observed that 194 of the 210 total virulence factor genes were present in nearly all (more than 98% of total strains; we may call them core genes) and 13 genes were not present at all in *H. pylori*-Mon strains (Appendix A). The remaining three genes, PARA protein gene HP1000 (136, 61.0%), *ceuE* (67, 30.0%), and *hcpB* (22, 9.9%), were highly diverse among *H. pylori*-Mon strains. The lowest distribution of the *ceuE* gene was found in patients with GC (*p* = 0.001) and a higher distribution in Sg2 strains (*p* < 0.001). While the other two genes did not show any significant variation depending on the diseases (Appendix A).

In addition, we analyzed OMP’s seven family genes in order to determine whether or not the genes in that family were associated with a particular gastric disease or subgroup at the presence or absence level. Similar to virulence factor genes, 50 out of 60 OMP genes were found to be core genes shared by more than 98% of the total strains except for the *bab* gene families (*hopS/babA*, *hopT/babB*, *hopU/babC*), *homD*, *hopP/sabA*, *hopO/sabB*, *horA*, *frpB2*, and *fecA*. As shown in Appendix A, Sg1 strains have more iron-regulating genes than Sg2 strains, such as the *frpB2* gene and the *fecA2* gene. In particular, *frpB2* was seen at a higher frequency in the GC patients. While *hopD* was less prevalent in the GC group and the Sg1 strains. Moreover, these findings were supported by the fact that, when we compared the average present genes of OMP by disease status and two main subgroups, it was interesting to note that iron-regulating OMP genes by disease status and total OMPs, Hop, Hom, and the iron-regulating OMP family, were significantly different between Sg1 and Sg2 strains (Figure 2A). In the next part, we will concentrate more on the *bab* families, *sabA,* and *sabB,* and their genotypes.

The association of 13 well-known virulence-associated genes presence or absence of gastric diseases was investigated further via an in-depth analysis that was based on the genotypes and “on” or “off” status. All virulence-associated genotypes of *H. pylori*-Mon strains are presented in Table 2.

The 170 (76.2%) *H. pylori*-Mon strains had a *cagA* gene, and 72.7% of those were Western-type CagA. Interestingly, all strains of Sg2 were *cagA*-negative, while *cagA* prevalence in Sg1 strains was 152/158 (96.2%) and mostly ABC-CagA (58.2% of them). About *vacA,* we typed *vacA* based on s-, i-, m-, c-, d-, and n- regions, and, generally, all strains have shown 22 different variations (Appendix A). We simplified the *vacA* genotype using s-, i-, and m- regions into five groups. Similar to CagA prevalence, 124 (55.6%) strains of *H. pylori*-Mon possessed the s1i1m1 allele type. In contrast, s1i1m1 (118, 74.7%) was also predominant in Sg1 strains, and it has been associated with GC cases more than other genotypes, while s2i3m2 *vacA* (37, 82.2%) was common in Sg2 strains. We conducted an investigation into the serine protease *htrA* S and L variants, distinguished by a substitution of serine (S) with leucine (L) at amino acid position 171 within the protease domain of the protein, with relevance to the development of GC. The L-variant of *htrA* was notably more prevalent among *H. pylori* strains obtained from Sg2 in comparison to those from Sg1 (*p* < 0.001). However, it is noteworthy that neither of these variants exhibited any significant disparity when comparing strains associated with different gastric diseases (Table 2). In addition, strains belonging to PUD and Sg1 had a higher frequency of *iceA1* compared to other strains. Conversely, Sg2 strains displayed a contrasting pattern, with a higher frequency of *iceA2B* and *iceA2C* genes than *iceA1* genes. Furthermore, a distinctive observation in GC strains was that the *iceA2D* gene exhibited the highest frequency compared to other strains.

The OMP genes *babA*, *babB*, *hopQ*, *hopZ*, *sabA*, *sabB*, and *homA/B* were included in the further investigation of gene genotypes that was carried out. The detected presence of genes including *babA*, *babB*, *sabA*, and *sabB* ranged from approximately 60.1% to 97.3% within the *H. pylori*-Mon strains (Table 2). Notably, the absence of the *babC* gene was observed in all strains, accentuating a distinct genetic pattern characterizing this specific strain set. We found that the *babA* gene had similar allele types in both Sg1 and Sg2 strains, and the AD4 allele was the most frequent (Appendix A). In the case of *babB,* its BD2 was more common in both Sg1 and Sg2 strains. Interestingly, unclassified *babB* genes were found in 33 (14.8%) strains of all strains during bioinformatic analysis. There were no significant differences between allele types of *babA* and *babB* between the diseases (Table 2). Table 3 shows the significant differences in *hopQ, homA/B* allele types with *sabA* and *sabB* presence between the *H. pylori* two subgroups, but not *hopZ*. Interestingly, *sabA* and *hopQ*-II were carried in all Sg2 strains, while *sabB* was absent. Inversely, we also found that 147 (93.0%) of Sg1 strains contained *hopQ*-I. From Hom family genes, we checked *homA/B* distribution, and, generally, *homA* and *homB* existed equally in all strains. However, the frequency of *homA* was significantly higher in Sg2 than Sg1 strains (Appendix A), while the frequency of *homB* was the inverse. The results revealed that the frequency of *hopQ*-I and *sabB* significantly increased in GC cases (both *p* < 0.001).

Next, we confirmed the “on” or “off” status for the *babB*, *hopZ*, *oipA*, *sabA*, and *sabB* genes and then analyzed their association with diseases and subgroups. In general, most of the strains contain “on” status for *oipA* (*n* = 176, 78.9%), *babB* (*n* = 121, 54.3%), and *sabA* (*n* = 116, 52.0%), while most of the strains contain “off” status for *hopZ* (*n* = 145, 65.0%), and *sabB* (*n* = 73, 32.7%) (Appendix A). Beginning with the “on/off” status association with disease, we found that *oipA* and *sabB* genes had significant associations between their “on” status and gastric disease (*p* < 0.001 and *p* = 0.003, respectively) (Figure 3). These associations especially appeared to be driven by the fact that strains that had “on” *oipA* and *sabB* were more likely (100% and 50.0%) to come from patients suffering from GC than patients with NAG and AG/IM (68.3–88.5% and 19.2–34.6%, Appendix A). It is also interesting to note the difference between the “on/off” state and the number of subgroups; we found that *oipA*, *babB*, and *sabB* “on” were significantly found in Sg1 compared to Sg2 (all *p* < 0.001). Conversely, *sabA* “on” was significantly found in Sg 2 compared to Sg1. Moreover, all strains from Sg2 did not possess any *sabB* genes.

### 3.3. Diversity of ICEHptfs in Gastric Cancer and Other Diseases

The T4SS is an essential membrane-spanning apparatus that is also the component of the *H. pylori* genome with the highest degree of plasticity. Along with looking at the virulence-associated genes, we also looked for the T4SS components: *cag*PAI (Cag T4SS), the ComB system, and the ICE*Hptfs*. The ICE*Hptfs* consist of ICE*Hptfs*3 (also known as *tfs3* T4SS) and ICE*Hptfs*4 (also known as *tfs4* T4SS). We determined each T4SS by the presence of the *xerT*, *virB2*, *virB3*, *virB4*, *virB6*, *virB7*, *virB8*, *virB9*, *virB10*, *virB11*, *virD2*, *virD4*, and *topA* genes. For *tfs4*, its cluster consists of two left (L1/2), central (C1/2), and right (R1/2) segments. The *tfs4* is subdivided into *tfs4a* (L2-C1-R2), *tfs4b* (L1-C1-R1), and *tfs4c* (L1/L2-C2-R2) according to the combinations of the segments that it contains.

The distribution of various types of T4SS components based on disease status is displayed in Table 3. Based on our results (Table 3 and Figure 4), the presence of *cag*PAI as well as *tfs3* and *tfs4* was determined in 172 (77.1%), 159 (71.3%), and 199 (89.2%) of *H. pylori*-Mon strains, respectively. Genes for the ComB system are present in every strain. Overall, it contained complete of *cag*PAI, had the most completed structure at 169 (75.8%), and other T4SS were 42 (18.8%) in *tfs3* and 61 (27.4%) in *tfs4*.

As was mentioned before, Sg1 and Sg2 strains have distinct differences in the pattern of disease outcomes that they cause in one another, as well as in the presence of certain virulence factors and genotypes. Previous research has shown that T4SSs, especially *cag*PAI and *tfs4*, have very clear traits that are directly related to their phylogeography and ancestral origins [30]. Therefore, we examined the T4SS gene diversity of Sg1 and Sg2 strains so that we could better understand the T4SS characteristics of each of these strains. In Figure 4 and Appendix A, the analysis clearly demonstrated that Sg1 strains had a higher frequency of all T4SS components, particularly *cag*PAI and *ctkA* genes, than Sg2 strains; however, Sg2 strains possessed significantly higher numbers of complete *tfs3* and *tfs4*, especially *tfs4b* type (L1-C1-R1 module) and *dupA* genes (all *p* < 0.001).

The AG/IM and GC groups showed the lowest frequency of *tfs4* completeness (*p* = 0.005), and inversely, almost all strains of GC possessed complete *cag*PAI. Among the strains analyzed, the presence and completeness of *tfs3* exhibited no notable differences based on the individuals’ disease status. With the exception of strains that incomplete *tfs4* (*tfs4*-other), strains displaying the *tfs4b* type (*n* = 31, 25.8%) were found to be more prevalent in patients with NAG compared to the *tfs4a* (*n* = 7, 5.8%) and *tfs4c* (*n* = 1, 0.8%) types. Furthermore, these *tfs4b* type strains also exhibited a tendency to be associated with NAG strains from Sg2. Notably, within *H. pylori*-Mon strains, a total of 126 strains (56.5%) were identified to possess all four T4SS components. Among these components, the combination of *cag*PAI and *tfs4* (*n* = 159, 71.3%) was observed more frequently than the combinations involving *cag*PAI and *tfs3*, or the complete *tfs4*. Moreover, it was observed that the occurrence of the *cag*PAI and *tfs4* combination was most prominent in strains from gastric cancer (GC) as compared to other disease groups.

Next, we examined the effector protein genes of *tfs3* and *tfs4*. The prevalences of *ctkA* and *dupA* alone were 37.2% and 25.1% of *H. pylori*-Mon, respectively. The *ctkA* was more frequently observed in PUD (55.6%), AG/IM (39.7%), and NAG (35.8%), but not in GC (25%) groups. The *dupA* alone was more frequent in the NAG (30.8%) group from Sg2 (Table 3 and Figure 4).

### 3.4. Risk Estimation of Significant Genes and Genotypes between Gastric Cancer and Other Diseases

We selected every significant virulence determinant, like genes and their genotypes or status (e.g., presence of *cagA*/*cag*PAI, *sabB*, complete *tfs4, tfs4b* type, *ceuE, hopD*, or *frpB2* gene; status of *vacA s1i1m1* allele, *hopQ*-I allele, *sabB* “on” and *oipA* “on”) linked to GC, in order to determine the relationships between each other, and which virulence factors had the most impact on gastric diseases. As a result of this analysis, a notable trend emerged; strains within the GC group exhibited a higher incidence of concurrent possession of *cagA*/*cag*PAI in conjunction with several other significant virulence determinants, including *vacA* s1i1m1 type, *hopQ* type I, *frpB2* gene, *oipA*, and *sabB* “on” status. This trend is visually illustrated in Appendix A, and further statistical assessments reveal positive correlations between these virulence factors.

Furthermore, we conducted a logistic regression analysis encompassing all the significant virulence determinants that are linked to gastric cancer. As a result of this, the strains that carry the *vacA*-s1i1m1 type and the *sabB* gene presence with the “on” status had a greater risk (OR = 2.9–5.1) for developing GC, and the *hopD* presence had a protective factor (OR = 0.27) for GC (Table 4).

### 3.5. Identification of SNPs and Motifs Associated with Gastric Cancer and Other Diseases by GWAS

In addition to the presence and genotype of genes, SNPs might play a significant role in the development of gastric carcinogenesis, particularly in the context of *H. pylori* infection [38,39]. Therefore, GWAS was conducted to examine significant SNPs in *H. pylori* strains linked to the GC phenotype. First, we selected the GWAS population only for inside Sg1 strains (*n* = 158) due to subgroup-specific genomic diversity, and then we did two-group comparisons such as GC (*n* = 15) vs. non-GC (*n* = 143, rest of strains except isolates from GC) at the SNP and k-mer level using a linear mixed model framework.

A total of 410,640 SNPs were called by Snippy, and 155,447 SNPs with a minor allele frequency greater than 5% were analyzed. As well as the total 2,601,024 k-mers called by the unitig-caller, 1,267,761 k-mers were analyzed. After analysis, we found significant SNPs (*n* = 19) and k-mers (*n* = 20) at a significant level of *p*-value, and we filtered them out according to criteria with allele frequency greater than 50% and less than 20% allele difference (Appendix A). Finally, we found the most significant nine non-synonymous and nine synonymous SNPs in seven and nine genes separately (Table 5 and Appendix A). Figure 5 shows the Manhattan plot of significant SNPs from the SNP-GWAS against the reference strain 26,695 genome. Risk genotypes exhibited a range of 33.3–40.0% allele frequency in the GC and 2.8–4.2% non-GC *H. pylori* strains.

The majority of significant SNPs were identified within the least known genes in terms of thorough functional explanations of *H. pylori* virulence, encompassing HP0452, HP1280 (genes encoding hypothetical proteins), HP1547 (*leuS* encoding leucyl-tRNA synthase), and HP1220 (*yhcG* encoding ABC transporter ATP-binding protein associated with multidrug-efflux proteins). In addition, three SNPs from all nine SNPs were located in the *hpyAV*, R-M system type II gene.

In Figure 6, we performed amino acid substitutions on predicted protein 3D structures to analyze their biological significance in relatively well-studied candidate genes, *ftsY* and *dld*-II.

The most significant SNP is found in the *ftsY* gene, which encodes the cell division protein, a functional homolog of the signal recognition particle (SRP) receptor, and is responsible for delivering the ribosome-nascent chain complex to the bacterial plasma membrane [40]. The amino acid substitution of *ftsY* SNP occurs at position 54 (leucine to serine) within the N domain, which consists of a four-helix bundle, and it may have the potential to induce minor conformational modifications (Figure 6A). Notably, this N domain, in conjunction with the G domain present in the FtsY and Ffh (SRP) proteins, plays a pivotal role in regulating the GTP hydrolysis mechanism, which is crucial for the co-translational targeting process [41].

In Figure 6B, the next interesting SNP corresponds an amino acid substitution at position 751 (valine to isoleucine) of the *dld*-II gene (HP1222) [42]. This gene encodes a putative D-lactate dehydrogenase-II (Dld-II), which is a NAD-independent enzyme that catalyzes the conversion of lactate to pyruvate [42]. The Val751Ile substitution occurs within the Fe-S binding domain of Dld-II. Such a modification may have the potential to impact the binding affinity of the respective site, subsequently exerting an influence on the growth or survival of *H. pylori*.

Within Appendix A, three out of the total of nine synonymous SNPs were ascertained within genes encoding hypothetical proteins (HP0463, HP0465, and HP0502). A point of particular interest arises from the observation that certain SNPs manifest a presence in two established virulence factors: specifically, *vapD*, responsible for encoding a virulence-associated protein D, and *faaA*, which encodes flagella-associated autotransporter A, representing constituents of the VacA-like protein family. Furthermore, an additional set of four SNPs was situated within essential genes: *hpyAXII*, responsible for encoding the adenine methyltransferase of the R-M system; *fadA*, responsible for encoding acetyl-CoA acetyltransferase; *rpsD*, responsible for encoding ribosomal protein S4; and *frdC*, responsible for encoding the fumarate reductase (FrdC). FrdC serves as the membrane anchor for the FrdA and FrdB proteins.

## 4. Discussion

*H. pylori* virulence factors play a pivotal role in carcinogenesis processes, and a wide range of virulence factor genes might modulate host cells in different ways, including inflammation, DNA damage, cell cycle regulation, and immune evasion, thereby promoting chronic gastritis to malignant transformation [43,44,45]. Here, we showed characterizations of gastric diseases, especially GC-associated *H. pylori* virulence determinants at the gene, their genotype, and SNP level, using abroad perspectives of virulence factor genes and the GWAS method. We have found several important findings. First, the *H. pylori*-Mon strains isolated from Mongolia have shown that there are two different pathogenic *H. pylori* strains causing different infection patterns even within a country: Sg1 strains that are highly virulent and Sg2 strains (hspAltai) that are less virulent in causing GC. Sg1 strains predominate in Mongolian patients, which is a high-risk population for GC. Second, Sg1 strains possess Western-type *cagA* with other virulence factor genotype genes such as *vacA* s1i1m1, *sabB* “on” status, *oipA* “on” status, and *hopQ*-I allele type. On the other hand, Sg2 strains do not have *cag*PAI genes, and they possess less virulent *vacA* s2i3m2, *sabB* absent or “off” status, *oipA* “off” status, and *hopQ*-II or I/II allele types. Third, *H. pylori*-Mon strains isolated from the same country had ICE*Hptfs* diversity, with complete ICE*Hptfs* being seen more in Sg2 (hspAtai) strains associated with a reduced risk of GC. Fourth, the majority of virulence factor genes carried by *H. pylori* are in the core genome, and these genes might affect host GC progression through their genetic variation at the SNP level. We found 18 novel candidate SNPs in 16 genes by SNP-GWAS and k-mer-GWAS methods.

A combination of host, bacterial, and environmental factors, including regional variations in *H. pylori* infection prevalence, affect the clinical implications of *H. pylori* infection and the incidence of GC. A significant factor in this context is the distribution of strains carrying the CagA protein and its subtypes, Western-type or East Asian-type CagA. Even though 76.2% of the *H. pylori* strains in this study were CagA-positive, it is important to note that, in the case of *H. pylori* strains from Mongolia in Sg1, the vast majority (96.2%) had Western-type CagA, which is a dominant characteristic in this population. CagA prevalence has been shown to vary considerably across geographic locations and it matches the prevalence of GC in previous research. For example, rates of 30–60% are common in North America, and Western European countries have a low rate of GC, whereas rates of 80–90% are common in East Asian countries classified as having a high risk of GC (particularly Japan, Korea, and China) [46,47]. In particular, our findings indicate that all Sg1 strains in Mongolia had Western-type CagA, accompanied by a complete *cag*PAI, which distinguishes them from other Western countries where isolated strains harbor Western-type CagA. This completeness and high prevalence of *cag*PAI may explain one of the reasons why Mongolia has the highest GC death and morbidity rates in the world. Importantly, our findings corroborate earlier research indicating that the presence of *cag*PAI alone is insufficient to explain the varying risk of GC across different populations. Rather, it is the interplay between host factors and other genetic determinants within the *H. pylori* genome that can adequately elucidate these distinct risk disparities [19,20,46]. For example, *H. pylori* that are *cagA*-positive also have *vacA* s1i1m1 status, *sabB* on status, *oipA* “on” status, and the *hopQ*-I allele, and these four additional-positive statuses may make *cagA*-positive strains more virulent than *cagA*-negative strains. Previous research has shown that, not only *vacA* s1 and m1, but also i1 and d1, may stimulate GC and PUD-related vacuolating activity in eukaryotic cells [48,49]. According to our findings, the *vacA* s1i1d1m1c1n1 type is the most prevalent in *H. pylori*-Mon (Appendix A). SabB is a putative adhesin that arose via a gene fusion involving the sialic acid-binding adhesin (SabA) gene [50]. However, very little is known about the role of this protein. A SabB, according to the findings of one study, may have a role in the changing gastric environment during the development of gastric disease by encouraging bacterial adhesin gene variation linked with increased colonization [50]. In a meta-analysis of 18 studies, “on” *oipA* status was associated with higher odds of PUD and GC [51]. This gene likely functions via the STAT1-IRF1-ISRE pathway and is required for complete activation of the IL-8 promotor [52]. IL-8 expression is linked to inflammation via neutrophils, tumorigenesis, and angiogenesis [53]. The outer membrane protein HopQ attaches to a molecule on the surface of gastric epithelial cells, called the carcinoembryonic antigen-related cell adhesion molecule, allowing the CagA protein to enter the cell [54]. *H. pylori* HopQ is a significant factor in GC because it aids in the import of the carcinogenic CagA protein into the cell. However, there are some controversial results from the *hopQ*-I and *hopQ*-II allele types association with GC [55]. More research with a larger sample size and from multiple centers is required. As an alternative approach to considering genotypes, a recent study has highlighted the substantial impact of SNPs on the development of GC. This is the specific substitution from “171S” to “171L” in the HtrA protein, which has been shown to enhance the protein’s proteolytic activity at the gastric epithelial junction, potentially amplifying the risk of cancer. This analysis performed the *htrA* genes in 1043 *H. pylori* genomes from worldwide sources (the majority are hpEastAsia strains from East Asian countries) [39]. However, our study indicates no discernible distinction in the presence of the 171L variant of HtrA among various gastric diseases, including GC, and the 171S variant was predominant in *H. pylori*-Mon strains. Consequently, it is important to note that the current genetic determinants, such as disease-related SNPs, exhibit variability contingent on distinct *H. pylori* phylogenetic groups. These variations emerge as a result of accumulated adaptive changes stemming from the coevolutionary relationship.

According to this study, *tfs3* and *tfs4* were found to be prevalent in 71.3% and 89.2% of *H. pylori*-Mon, respectively, which is comparable with Western countries but higher than East Asian countries; for example, *tfs3* was found to be 58% prevalent in the United States, 45% prevalent in Vietnam, and 54.3% prevalent in Indonesia. [56,57,58,59]. The prevalence of *dupA* among *H. pylori*-Mon strains was 25.1%, which is lower than the prevalence in both Western (69.3% in the USA) and East Asian (37.0% in Japan, 37.0% in Korea, 37.0% in Colombia, 55.0% in Vietnam, and 15% in Indonesia) countries [56,57,58,59]. In line with prior research, we found no evidence that *dupA* affects disease outcomes. In addition, no GC instances can be represented by the presence of *dupA* with complete *tfs4* and *tfs4b* types. Interestingly, we found a higher frequency of *ctkA* (37.2%) among *H. pylori*-Mon strains than many other countries, such as the USA (2%), Cambodia (25%), and Vietnam (29%). This *ctkA* encodes cell translocating kinase A, which increases host cell contact of the pro-inflammatory protein, upregulates NF-kappaB signaling, and induces TNF-alpha and IL-8 cytokine production from cultured macrophages, suggesting that it may enhance the *H. pylori*-mediated inflammatory response [60]. Despite no evidence of a significant association with disease outcomes, the presence of *ctkA* may explain the persistent inflammatory process and atrophy in Sg1 strains. When looked at from an alternative perspective, the *tfs3* and *tfs4* clusters in the Sg1 and Sg2 strains were distinct from one another. Complete *tfs3* and *tfs4b* types (L1-C1-R1) are found commonly in Sg2 strains (hspAltai, closely related to hspIndigenousAmericas strains), whereas Sg1 strains are often incomplete *tfs4* type (L2-C1-R1 or incomplete). According to previous research, ancestral populations like hpAfrica2 are more likely to have the *tfs4b* gene [30]. However, hpAfrica1, hpEastAsia, and hpEurope commonly carry the *tfs4* type, which is itself a combination of the *tfs4b* and *tfs4c* (L2-C2-R2) types.

In the present study, we used GWAS to evaluate the genomic differences between Sg1 *H. pylori* strains originating from non-GC hosts (*n* = 143) and from GC hosts (*n* = 15). Under this comparison, *H. pylori* with a certain genetic variant may advance to a GC that is distinct from the same pathogenicity cascade as NAG. Previously, two GWAS studies of GC were published [22,23]. The first GWAS was conducted on a 173 hpEurope *H. pylori* strain from 124 non-GC (NAG and AG/IM) and 49 GC cases. They identified 32 significant SNPs, and most of them are located in the *cag*PAI, such *as cagT*, *cagX*, *cagH*, *cagI*, *cagE*, *cagU*, *cagβ*, cag7, some OMP genes such as *babA*, *omp27*, HP1055, and other genes (*hpaA*, *ligA*, *ychF*, *rimM*, *dnaE*, *azlC*, etc.). A second GWAS was conducted on hspEAsia 240 *H. pylori* strains from 115 duodenal ulcers (DU) and 125 GC cases. This group determined 11 significant SNPs, 3 DNA motifs, several OMP genes (*fecA2*, *ompA101, frpB2*), an R-M system gene (*hsdM*), several enzymes (*isp*, *rpoZ*, *recG*), and other genes, such as, *trkA*, *dsbG*, *triH*, *tipC*, *thiE*, and *csd5*. Interestingly, no SNPs matched between the two studies. It may be explained by differences in control groups (NAG, AG/IM, or DU) and/or subpopulation-specific *H. pylori* genetic variance (hpEurope or hspEAsia). The GWAS in this study is partially similar to the latter GWAS. We found genes with striking functional similarities. The *hpyAV* (HP0053), which is one member of type II R-M systems, and the *dld*-II (HP1222). A *dld*-II, a gene encoding D-lactate dehydrogenase (D-LDH), shares a functional group with *lldF* and *tlpC*, the significant genes from the GWAS of hspEAsia *H. pylori* strains. The product of the HP1222 gene is a D-LDH that does not need NAD to operate and also makes a contribution to the oxidation of L-lactate [61]. Lactate is used in the energy cycle and the production of pathogenic determinants such as lipopolysaccharide and polysialic acid capsules in those bacteria [62]. Without lactate, bacteria cannot colonize and live in vivo [62]. Intriguingly, the epithelial cells of the gastric antrum release notably higher quantities of lactate in comparison to the corpus, and it is noteworthy that *H. pylori* substantially depletes this available lactate [63]. The *tlpC* is a chemotaxis receptor and is responsible for lactate binding proteins. Furthermore, our findings posit the possibility of potential virulence-associated genes emerging from the analysis of synonymous SNPs, as exemplified by genes like *faaA* and *vapD*. Both of these genes contribute to the augmentation of *H. pylori*’s capability to establish stomach colonization in vivo [64,65]. Specifically, *faaA* is associated with flagellar localization [66]. The *faaA* mutant revealed instances of flagellar mislocalization, accompanied by a reduction in bacterial motility [66]. The *vapD* gene plays a pivotal role in facilitating the enduring presence of *H. pylori* within gastric cells over a long period of time, thereby substantively contributing to the progression of progressive, severe inflammations and conceivably inciting a cascade that potentially leads to the initiation of a carcinogenic process within the gastric epithelial cells [65]. To gain a more precise explanation of the other hypothetical gene, further experimental research is needed to understand these gene expression patterns, and their interactions with other molecules or proteins affecting gastric carcinogenesis.

Moreover, we found one OMP gene, *frpB2* (Appendix A), had significantly higher prevalence in *H. pylori* of GC cases, and *fecA2* had a lower prevalence in Sg2 strains based on gene presence and absence analysis. *H. pylori* may sustain its life and increase its virulence by collecting iron [67]. There are numerous intriguing characteristics of iron-regulated outer membrane proteins and their association with GC. During infection, *H. pylori* has developed complicated ways to scavenge iron from the host [67]. One of the bacteria’s methods is the development of iron regulated OMPs (FecA2, FrpB2, etc.) [15]. This implies a possible relationship between iron acquisition and GC development.

There are some limitations in this study that are worth mentioning for future studies. In the beginning, the total number of GC cases in this study was limited to 16 cases, despite Mongolia’s high prevalence of GC. A future study with a higher number of strains isolated from GC patients is recommended, although we tackle this limitation by increasing the strength of the statistical tests and the generalizability of the results. The genetic variation and connections within the *H. pylori* population are possibly more complex than this analysis using current data suggests. It is worth mentioning that we collected gastric biopsies from about a hundred GC patients. Unfortunately, not so many could be cultivated. *H. pylori* might be eliminated from the malignant mucosa in a “natural way”. Previously, our study found that our investigation using 16S rRNA microbiome analysis confirmed this intriguing phenomenon [68]. However, as a first report and pilot investigation, we used the Sg1 predominant subgroup in Mongolia, which is located in Northeast Asia. Our findings using the Sg1 subgroup, imply that GWAS should be conducted separately for each *H. pylori* subgroup because of their inherent genetic diversity. To overcome this limitation and better understand *H. pylori* genetic determinants, further research with bigger and more diversified sample numbers, complete genomic coverage, and rigorous study methods is needed.

## 5. Conclusions

In summary, through the application of population genomics and a comprehensive analysis of virulence-associated genes, our study reveals that highly virulent strains of *H. pylori* exhibit the capability to accumulate multiple variations as an adaptive strategy that is influenced by variations in the host. This study highlights the knowledge of genomic variation in highly virulent *H. pylori* in the region with the highest GC incidence. The comprehension of the occurrence of virulence factors in *H. pylori* in gastric diseases from a broad perspective, including the wide range of virulence-associated genes and a novel candidate by GWAS, is critical since all the factors may be related to the carcinogenesis mechanism in a particular way. Further studies are needed to investigate the functional characteristics of these genes and their contribution to *H. pylori* pathogenesis, providing insights into the development of strategies to combat *H. pylori*-associated GC.

## Figures and Tables

**Figure 1 cancers-15-04528-f001:**
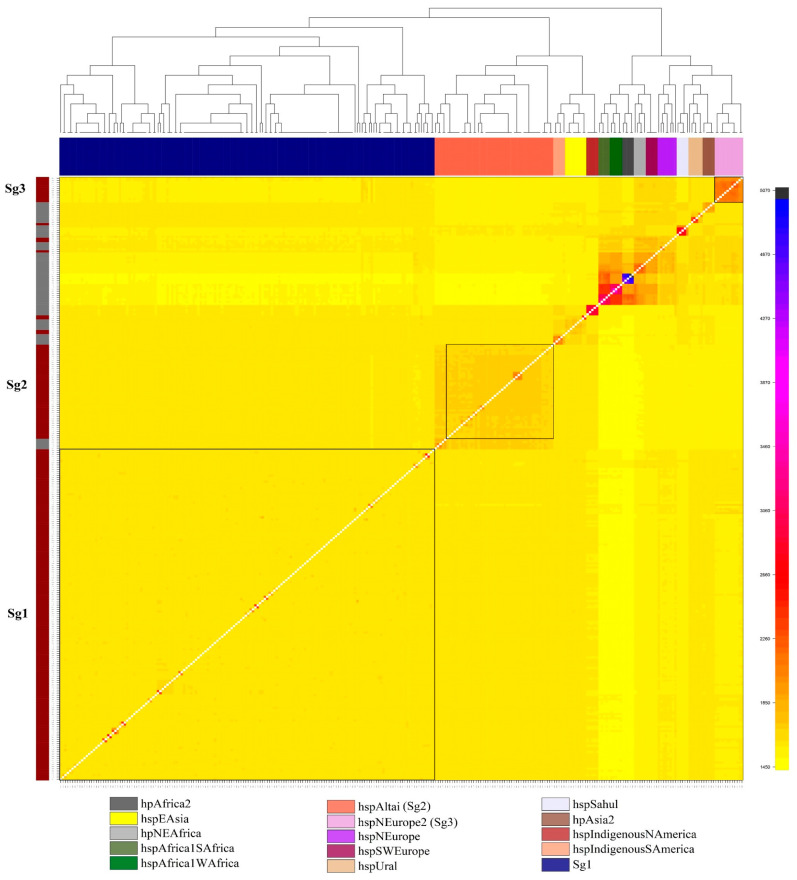
Population structure of *H. pylori*-Mon strains with reference strains. Matrix cells are color-coded according to the estimated chunks of DNA transferred from donors (*y*-axis) to recipient genomes (*x*-axis). The *H. pylori*-Mon strain is represented by a dark red hue on the left color strip, whereas other reference strains are represented by a gray color. The *H. pylori* subgroups were represented by the top color bar.

**Figure 2 cancers-15-04528-f002:**
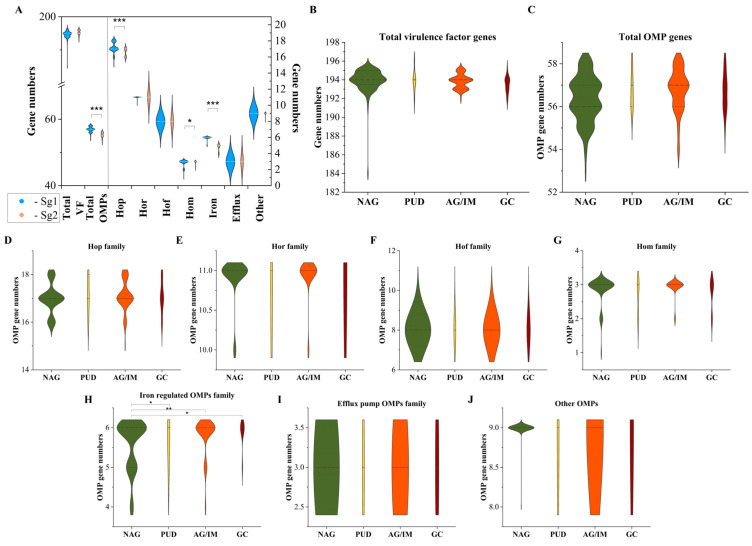
Analysis of a set of virulence factors and OMP family genes in disease and *H. pylori* subgroups. We compared the median virulence factor and OMP gamily genes between two main *H. pylori* subgroups (**A**). The median of virulence factor (**B**) and OMP family genes (**C**) were 195 and 57 genes were variable along with Hop family genes (**D**), Hor family genes (**E**), Hof family genes (**F**), Hom family genes (**G**), iron-regulating OMP family genes (**H**), efflux pump OMP family genes (**I**), and other OMP genes (**J**) in the disease groups. Statistical significances between *H. pylori* subgroups and disease groups were assessed by Mann–Whitney U test and Kruskal–Wallis test: *-*p* value < 0.05, **-*p* value < 0.01, ***-*p* value < 0.001. Sg1 = subgroup 1, Sg2 = subgroup 2, OMP = outer membrane protein, NAG = non-atrophic gastritis, PUD = peptic ulcer disease, AG/IM = atrophic gastritis/intestinal metaplasia, GC = gastric cancer.

**Figure 3 cancers-15-04528-f003:**
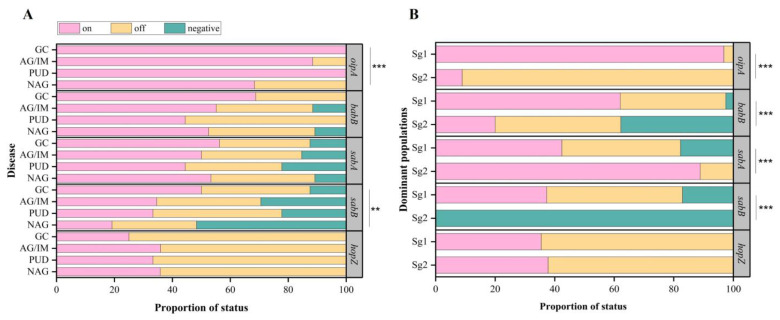
*H. pylori* known OMP functional status by disease status (**A**) and *H. pylori* subgroups (**B**). The “on” or “off” status was evaluated CT-repeat pattern using a nucleotide sequence of each gene defined by corresponding previous studies. **- *p* value < 0.01, ***-*p* value < 0.001. Sg1 = subgroup 1, Sg2 = subgroup 2, NAG = non-atrophic gastritis, PUD = peptic ulcer disease, AG/IM = atrophic gastritis/intestinal metaplasia, GC = gastric cancer.

**Figure 4 cancers-15-04528-f004:**
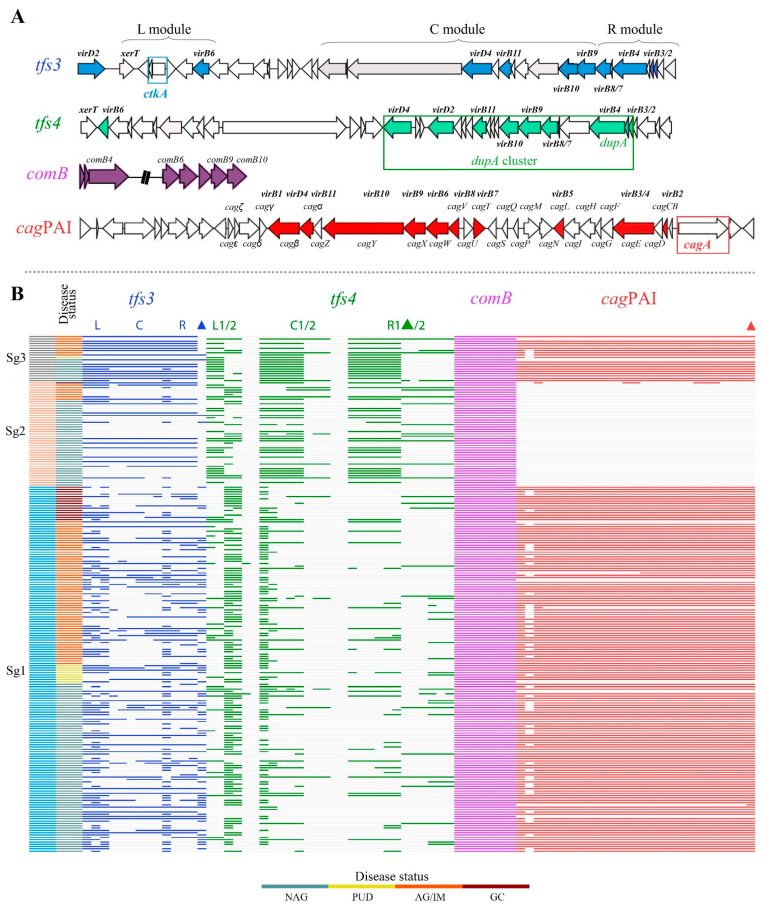
Gene content comparison of T4SS components by disease status and *H. pylori* populations. (**A**) The *tfs3, tfs4, comB*, and *cag*PAI gene arrangements in reference strains Gambia 94/24, P12, and 26695. In each T4SS component, colored genes were *vir* homolog T4SS genes. (**B**) The integrity of ICE*Hptfs* with *comB* and *cag*PAI genes is shown by a heatmap. The color strip on the left represents *H. pylori* subgroups and disease status. Following that, *tfs3* (blue), *tfs4* (green), *comB* (pink), and *cag*PAI (red) were colored by ortholog genes found inside (L-left ½, C-central ½, R-right ½) segments. Each gene’s presence is denoted by a different color triangle: *ctkaA* (blue, *tfs3*), *dupA* (green, *tfs4*), and *cagA* (red, *cag*PAI). Sg1 = subgroup 1, Sg2 = subgroup 2, OMP = outer membrane protein, NAG = non-atrophic gastritis, PUD = peptic ulcer disease, AG/IM = atrophic gastritis/intestinal metaplasia, GC = gastric cancer.

**Figure 5 cancers-15-04528-f005:**
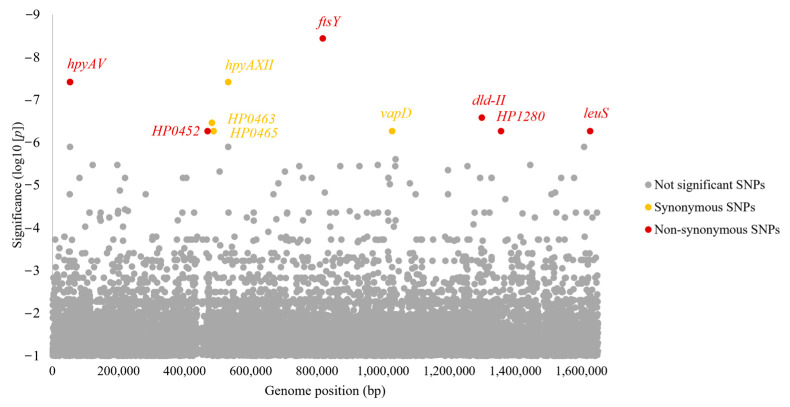
A significant variation in multiple genes related to Sg1 *H. pylori* strains from GC was identified using SNP-GWAS. The Manhattan plot displays negative log10-transformed *p* values (derived using the GWAS likelihood-ratio test). There are significant non-synonymous SNPs (red dots) and synonymous SNPs (yellow dots).

**Figure 6 cancers-15-04528-f006:**
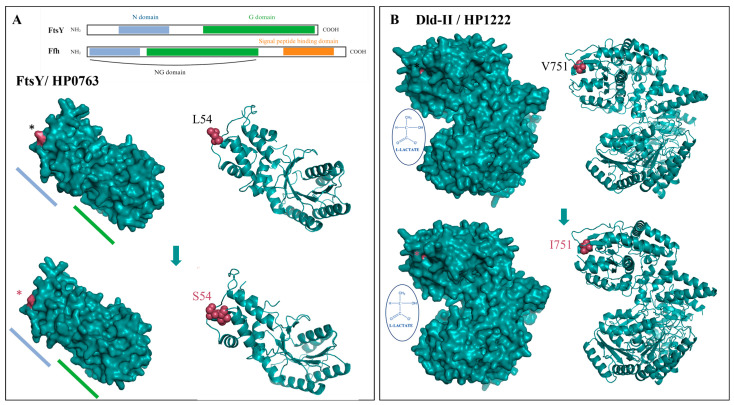
The anticipated 3D protein structures featuring significant non-synonymous SNPs are presented as follows: (**A**) the model showcases the de novo structure, highlighting the leucine and the mutated serine at position 54 (*) of the 26,695 strain FtsY/HP0763 protein. This visualization was conducted in silico using PyMOL. (**B**) The model offers insight into de novo structure, illustrating the valine and the mutated isoleucine at position 751 (*) of the 26,695 strain Dld-II/HP1222 protein. This representation was generated in silico and visualized using PyMOL.

**Table 1 cancers-15-04528-t001:** The disease status among the *H. pylori* populations.

*H. pylori.* Populations	Non-GC Group			GC group (*n* = 16)	OR ^a^ (95% CI)	*p* ^b^
	NAG (*n* = 120)	PUD (*n* = 9)	AG/IM (*n* = 78)			
Sg1 (*n* = 158)	73 (60.8%)	8 (88.9%)	62 (79.5%)	15 (93.8%)	6.7 (0.9–51.9)	0.044
Other (*n* = 65)	47 (39.2%)	1 (11.1%)	16 (20.5%)	1 (6.2%)		

^a^ Odds ratio for GC between Sg1strains and other groups. ^b^ Fisher’s exact test. No significance by logistic regression test.

**Table 2 cancers-15-04528-t002:** Genotyping of selected OMP and virulence factor genes and disease status.

Genes	Genotypes	NAG (*n* = 120)	PUD (*n* = 9)	AG/IM (*n* = 78)	GC (*n* = 16)	Total (*n* = 223)	*p* ^a^
*cagA*	AB ^1^	4 (3.3%)	0	0	0	4 (1.8%)	0.006
	ABC ^2^	49 (40.8%)	3 (33.3%)	43 (55.1%)	8 (50.0%)	103 (46.2%)	
	ABCn ^3^	25 (20.8%)	4 (44.4%)	19 (24.4%)	7 (43.8%)	55 (24.7%)	
	ABD ^4^	2 (1.7%)	1 (11.1%)	5 (6.4%)	0	8 (3.6%)	
	Negative	40 (33.3%)	1 (11.1%) ^b^	11 (14.1%) ^b^	1 (6.3%) ^b^	53 (23.8%)	
*vacA*	s1i1m1	52 (43.3%)	6 (66.7%) ^b^	53 (67.9%) ^b^	13 (81.3%) ^b^	124 (55.6%)	0.014
	s1i1m2	12 (10.0%)	1 (11.1%)	4 (5.1%)	1 (6.3%)	18 (8.1%)	
	s1i3m2	19 (15.8%)	1 (11.1%)	10 (12.8%)	1 (6.3%)	31 (13.9%)	
	s2i3m2	31 (25.8%)	0 ^b^	7 (9.0%) ^b^	1 (6.3%) ^b^	39 (17.5%)	
	other	6 (5.0%)	1 (11.1%)	4 (5.1%)	0	11 (4.9%)	
*htrA*	S variant	96 (80.0%)	9 (100%)	69 (88.5%)	14 (87.5%)	188 (84.3%)	0.118
	L variant	24 (20.0%)	0	9 (11.5%)	2 (12.5%)	35 (15.7%)	
*iceA*	Type 1	57 (47.5%)	7 (77.8%)	41 (52.6%)	4 (25.0%)	109 (48.9%)	0.055
	Type 2A	2 (1.7%)	0	4 (5.1%)	2 (12.5%)	8 (3.6%)	
	Type 2B	16 (13.3%)	0	7 (9.0%)	0	23 (10.3%)	
	Type 2C	20 (16.7%)	1 (11.1%)	9 (11.5%)	1 (6.3%)	31 (13.9%)	
	Type 2D	24 (20.0%)	1 (11.1%) ^b^	17 (21.8%)	9 (56.3%) ^b^	51 (22.9%)	
	Negative	1 (0.8%)	0	0	0	1 (0.4%)	
*babA*	AD2	2 (1.7%)	0	1 (1.3%)	0	3 (1.3%)	0.417
	AD3	36 (30.0%)	5 (55.6%)	25 (32.1%)	4 (25.0%)	70 (31.4%)	
	AD4	77 (64.2%)	4 (44.4%)	52 (66.7%)	11 (68.8%)	144 (64.6%)	
	Negative	5 (4.2%)	0	0	1 (6.3%)	6 (2.7%)	
*babB*	BD1	22 (18.3%)	2 (22.2%)	19 (24.4%)	5 (31.3%)	48 (21.5%)	0.516
	BD2	46 (38.3%)	4 (44.4%)	30 (38.5%)	9 (56.3%)	89 (39.9%)	
	BD3	19 (15.8%)	2 (22.2%)	10 (12.8%)	1 (6.3%)	32 (14.3%)	
	unclassified	21 (17.5%)	1 (11.1%)	10 (12.8%)	1 (6.3%)	33 (14.8%)	
	Negative	12 (10.0%)	0	9 (11.5%)	0	21 (9.4%)	
*hopQ*	Type I	74 (61.7%)	8 (88.9%) ^b^	65 (83.3%) ^b^	15 (93.8%) ^b^	162 (72.6%)	<0.001
	Type II	42 (35.0%)	0 ^b^	9 (11.5%) ^b^	1 (6.3%) ^b^	52 (23.3%)	
	Type I/II	4 (3.3%)	1 (11.1%)	4 (5.1%)	0	9 (4.0%)	
*hopZ*	Type I	97 (80.8%)	8 (88.9%)	73 (93.6%)	14 (87.5%)	192 (86.1%)	0.121
	Type II	23 (19.2%)	1 (11.1%)	5 (6.4%)	2 (12.5%)	31 (13.9%)	
*sabA*	Present	107 (89.2%)	7 (77.8%)	66 (84.6%)	14 (87.5%)	194 (87.0%)	0.688
	Negative	13 (10.8%)	2 (22.2%)	12 (15.4%)	2 (12.5%)	29 (13.0%)	
*sabB*	Present	58 (48.3%)	7 (77.8%) ^b^	55 (70.5%) ^b^	14 (87.5%) ^b^	134 (60.1%)	0.001
	Negative	62 (51.7%)	2 (22.2%) ^b^	23 (29.5%) ^b^	2 (12.5%) ^b^	89 (39.9%)	
*homA/B*	*homA*	67 (55.8%)	3 (33.3%) ^b^	32 (41.0%) ^b^	3 (18.8%) ^b^	105 (47.1%)	0.115
	*homB*	45 (37.5%)	6 (66.7%)	41 (52.6%)	10 (62.5%)	102 (45.7%)	
	*homA/B*	5 (4.2%)	0	3 (3.8%)	2 (12.5%)	10 (4.5%)	
	*homI*	3 (2.5%)	0	2 (2.6%)	1 (6.3%)	6 (2.7%)	

^1^ AB (AB *n* = 3, ABBB *n* = 1). ^2^ ABC (ABC *n* = 101, AABC *n* = 1, ABBBC *n* = 1). ^3^ ABCn (ABCC *n* = 52, ABCCC *n* = 3). ^4^ ABD (ABD *n* = 4, AABD *n* = 1, ABBD *n* = 2, ABDC *n* = 1). ^a^ Global *p* value between disease groups was assessed by Chi-Square test. ^b^ Adjusted *p* < 0.05 versus NAG group was determined by Fisher’s exact test (Bonferroni method). NAG = non-atrophic gastritis, PUD = peptic ulcer disease, AG/IM = atrophic gastritis/intestinal metaplasia, GC = gastric cancer.

**Table 3 cancers-15-04528-t003:** Distribution of T4SS components by disease status.

T4SS	Genotypes			NAG (*n* = 120)	PUD (*n* = 9)	AG/IM (*n* = 78)	GC (*n* = 16)	Total (*n* = 223)	*p* ^a^
*cag*PAI	Presence ^§^		Y	80 (66.7%)	8 (88.9%) ^b^	68 (87.2%) ^b^	16 (100.0%) ^b^	172 (77.1%)	<0.001
			N	40 (33.3%)	1 (11.1%) ^b^	10 (12.8%) ^b^	0 (0.0%) ^b^	51 (22.9%)	
	Completeness ^§§^		Y	79 (65.8%)	8 (88.9%) ^b^	67 (85.9%) ^b^	15 (93.8%) ^b^	169 (75.8%)	<0.001
			N	1 (0.8%)	0 ^b^	1 (1.3%) ^b^	1 (6.3%) ^b^	3 (1.3%)	
	*cagA* presence		Y	80 (66.7%)	8 (88.9%) ^b^	67 (85.9%) ^b^	15 (93.7%) ^b^	170 (98.8%)	0.274
			N	40 (33.3%)	1 (11.1%) ^b^	11 (14.1%) ^b^	1 (6.3%) ^b^	2 (1.2%)	
*comB*	Presence ^§^ and Completeness ^§§^		Y	120 (100%)	9 (100%)	78 (100%)	16 (100%)	223 (100%)	-
			N	0	0	0	0	0	
*tfs3*	Presence ^§^		Y	84 (70.0%)	7 (77.8%)	57 (73.1%)	11 (68.8%)	159 (71.3%)	0.927
			N	36 (30.0%)	2 (22.2%)	21 (26.9%)	5 (31.3%)	64 (28.7%)	
	Completeness ^§§^		Y	28 (23.3%)	1 (11.1%)	12 (15.4%)	1 (6.3%)	42 (18.8%)	0.132
			N	56 (46.7%)	6 (66.7%)	45 (57.7%)	10 (62.5%)	117 (52.5%)	
	*ctkA*		Y	43 (35.8%)	5 (55.6%)	31 (39.7%)	4 (25.0%)	83 (37.2%)	0.451
			N	77 (64.2%)	4 (44.4%)	47 (60.3%)	12 (75.0%)	140 (62.8%)	
*tfs4*	Presence ^§^		Y	101 (84.2%)	8 (88.9%)	74 (94.9%)	16 (100.0%) ^b^	199 (89.2%)	0.022
			N	19 (15.8%)	1 (11.1%)	4 (5.1%)	0 ^b^	24 (10.8%)	
	Completeness ^§§^		Y	42 (35.0%)	1 (11.1%)	16 (20.5%) ^b^	2 (12.5%) ^b^	61 (27.4%)	0.005
			N	59 (49.2%)	7 (77.8%)	58 (74.4%) ^b^	14 (87.5%) ^b^	138 (61.9%)	
	*dupA*		Y	37 (30.8%)	2 (22.2%)	15 (19.2%)	2 (12.5%)	56 (25.1%)	0.163
			N	83 (69.2%)	7 (77.8%)	63 (80.8%)	14 (87.5%)	167 (74.9%)	
	Type	*tfs4a*		7 (5.8%)	0	1 (1.3%)	1 (6.3%)	9 (4.0%)	0.012
		*tfs4b*		31 (25.8%)	1 (11.1%)	12 (15.4%)	1 (6.3%)	45 (20.2%)	
		*tfs4c*		1 (0.8%)	0	2 (2.6%)	1 (6.3%)	4 (1.8%)	
		*tfs4*-other ^§§§^		62 (51.7%)	7 (77.8%) ^b^	59 (75.6%) ^b^	13 (81.3%) ^b^	141 (63.2%)	
Combinations	*cag*PAI+*tfs3*			69 (57.5%)	6 (66.7%)	51 (62.4%)	11 (68.8%)	137 (61.4%)	0.224
	*cag*PAI+*tfs4*			72 (60.0%)	7 (77.8%)	64 (82.1%)	16 (100%)	159 (71.3%)	0.229
	*tfs3*+*tfs4*			75 (62.5%)	6 (66.7%)	54 (69.2%)	11 (68.8%)	146 (65.5%)	0.313
	*tfs3*+*tfs4a*			2 (1.7%)	0	1 (1.3%)	0	3 (1.3%)	0.386
	*tfs3*+*tfs4b*			19 (15.8%)	0	10 (12.8%)	1 (6.3%)	30 (7.4%)	
	*tfs3*+*tfs4c*			0	0	0	0	0	
	All T4SS			62 (51.7%)	5 (55.6%)	48 (61.5%)	11 (68.8%)	126 (56.5%)	0.398

^§^ The presence of the T4SS was established by the assessment of the existence of either one of the following genes: *xerT*, *virB2*, *virB3*, *virB4*, *virB6*, *virB7*, *virB8*, *virB9*, *virB10*, *virB11*, *virD2*, *virD4*, and *topA*. For *cag*PAI, presence was established by the assessment of the existence of either one of the following genes: *virB1*, *virB2*, *virB3*, *virB4*, *virB6*, *virB7*, *virB8*, *virB9*, *virB10*, *virB11*, and *virD4*. ^§§^ The presence of all *virB2*, *virB3*, *virB4*, *virB6*, (in the case of *cag*PAI, *virB7* is included), *virB8*, *virB9*, *virB10*, *virB11*, *virD2*, and *virD4* genes were used to determine the level of completeness of each T4SS. ^§§§^ Unclassified and incomplete *tfs4* type. ^a^ Global *p* value between disease groups was assessed by Chi-Square test. ^b^ Adjusted *p* < 0.05 versus NAG group was determined by Fisher’s exact test (Bonferroni method). NAG = non-atrophic gastritis, PUD = peptic ulcer disease, AG/IM = atrophic gastritis/intestinal metaplasia, GC = gastric cancer. Y: yes, N: no.

**Table 4 cancers-15-04528-t004:** Age- and sex-adjusted risk for the GC in relation to significant genes and their genotypes or functional status in *H. pylori*-Mon strains by univariate logistic regression analysis.

	Univariate Logistic Regression		
	OR ^a^	OR 95% CI	*p*
*cagA* **presence**/*oipA* **“on”** vs. *cagA* absence/*oipA* “off” status	5.0	0.6–39.0	0.122
*vacA* **s1i1m1** vs**.** vacA other alleles	3.7	1.0–13.5	0.044
*hopQ*-**I** vs. II or I/II alleles	6.1	0.8–47.4	0.083
*sabB* **presence** vs. absence	5.1	1.1–22.9	0.035
*sabB* **“on”** vs. “off” status	2.9	1.0–8.1	0.042
***tfs4* complete** vs. incomplete	0.4	0.1–1.6	0.183
***tfs4b*** vs. *tfs4* other types	0.3	0.0–1.9	0.182
*ceuE* **presence** vs. absence	0.14	0.0–1.1	0.062
*hopD* **presence** vs. absence	0.27	0.1–0.8	0.014
*frpB2* **presence** vs. absence	4.3	0.6–33.3	0.164
Iron regulated OMP genes (numbers) **many** vs. less	2.3	0.6–8.9	0.240

^a^ Odds ratio for GC. Risk variables are represented in bold.

**Table 5 cancers-15-04528-t005:** Significant non-synonymous SNPs and k-mers for GC were determined in genome-wide association studies of 158 *H. pylori* isolates from Sg1 strains.

Types	*p*	Gene Locus in 26,695	Gene Name	Position in the 26,695 Genome	Position in the Gene	Safe Genotype	Risk Genotype	Frequency of Non-GC/GC, %	Effect on Amino Acid Sequence	Predicted Function
SNP	3.62 × 10^−9^	HP0763	*ftsY*	817,043	161	T	C	2.8/40.0	Leu54Ser	Signal recognition particle-docking protein
SNP	3.80 × 10^−8^	HP0053	*hpyAV*	52,671	1048	G	A	3.5/40.0	Leu350Phe	R-M system type II
SNP	3.80 × 10^−8^			52,679	1039	G	T	3.5/40.0	Thr347Asn	
SNP	3.80 × 10^−8^			52,681	1041	C	T	3.5/40.0		
SNP	2.57 × 10^−7^	HP1222	*dld*-II	1,297,719	2251	C	T	4.2/40.0	Val751Ile	FAD-binding and (Fe-S)-binding domain-containing protein
SNP	5.39 × 10^−7^	HP0452	hyp ^a^	469,160	405	G	T	2.8/33.3	Glu135Asp	(McrB family protein)
SNP	5.39 × 10^−7^	HP1280	hyp ^a^	1,355,874	808	C	T	2.8/33.3	Thr270Val	(Anthranilate phosphoribosyltransferase)
SNP	5.39 × 10^−7^	HP1547	*leuS* ^a^	1,625,904	2014	C	T	2.8/33.3	Ala672Thr	Leuci-tRNA ligase
k-mer	3.62 × 10^−9^	HP1220	*yhcG*	1,295,987–1,296,020	283–316	other ^b^	TTTTACAAGGATTTTTTTAGCGATTTTGATCCAT	2.8/40.0	multiple	ABC transporter ATP-binding protein

^a^ Hypothetical protein gene. ^b^ The other motifs than the risk genotype-motif considered as safe genotype.

## Data Availability

All extracted OMP genes from genotyping analysis are available upon request.

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
