# Peer review of "Study of Helicobacter pylori Isolated from a High-Gastric-Cancer-Risk Population: Unveiling the Comprehensive Analysis of Virulence-Associated Genes including Secretion Systems, and Genome-Wide Association Study"

_cancers, 2023, doi:10.3390/cancers15184528_

Round 1

Reviewer 1 Report

The authors present a genomic analysis of several recently isolated Helicobacter pylori strains from a geographic region at high risk for gastric cancer. Specifically, multiple virulence factors were characterized for each strain in terms of sequence conservation, completeness, allelic variants, SNPs and probable functionality; factors are considered as single genes or grouped into functional systems (cag, t4ss, t3ss). All these data are compared with different pathological conditions of bacteria-infected gastric tissue, from mild gastritis to gastric cancer. The analysis seeks to highlight the specific factors related to the risk of developing gastric cancer, whether they are favorable to cancer development (pro-tumorigenic) or negatively correlated with this disease (protective effect).

The topic is interesting, the study considers many bacterial genes, and some conclusions seem solid. However, much information is missing, some tables seem to contain incorrectly reported data, some data are reported as difficult to read, and some concepts are not sufficiently supported by the analysis. All these points need to be addressed before publication.

Text:

-          First of all, I have some problems with the word “secretome”: many virulence factors considered in the analysis are non-secreted outer membrane proteins, inner membrane proteins or cytoplasmatic factors (arsR). Although some outer membrane proteins are present in the OMVs, it is questionable that they can be considered part of the secretome. Non-secreted cytoplasmatic factors cannot be considered part of the secretome

-          Secondly, the inclusion of “hypothetical proteins” or similar genes in the panel of virulence factor is questionable. If they are enlisted in the VFDB repository, add a comment in the text

- The text is rich of tables and the same datas are reported both with a sg1/sg2 subdivision and the clinical class subdivision. Consider to leave only the table with clinical subdivision, which is the focus of the paper.

- As GWAS applied to different cohorts gives back completely different results, what is the significance and the scientific validity of this analysis in the manuscript?

-          All the acronyms must be explained (Mgl?)

-          Line 431: it is unclear the meaning of “30.5-37.2% allele frequency”. In Table 6 “Frequency, % Non-GC/GC” seems to be interpreted as the ratio of frequency between non-GC and GC. Example for HP0763 is reported 2.8/40.0: how these number fits with the sentence above? Explain better both text and table

-          Line 112: TFSS should be converted in T4SS

-          Line 286-287: it is unclear who are the 10 genes and globally the phrase has not significance

-          Line 282-296: Fig 2 panels B to J should be accurately commented in the text

-          Line 315: explain the rationale of the choice

-          Line 316-317: the sentence “Nearly all strains had 217 babA (97.3%) and 202 babB (90.6%) genes” has no significance in this form

-          Line 334-344: include details of how this analysis was performed in m&m, add a supplementary table with results and explain well in the text the rationale and the significance of the analysis, not only the comments of the % of on/off

-          Line 398-400: “The tfs4b cluster was found to be present more often than the other two clusters, and the vast majority of it was found in NAG strains (31, 25.8%).” The reported number and % is meaningless. If authors sustain that tfs4b cluster is more present in NAG strain (and it is true), the percentage should be calculated throughout the row, comparing this cluster in the different clinical classes

-          Line 438-448: non synonymous mutations with differential association with GC are not considered, nor commented. Please add a comment on their significance. If they are not significant, the analysis is still scientifically reliable?

-          Line 446: I don’t understand the significance of the phrase “HP1547 (leuS) is leucyl-tRNA synthase in bacterial cytoplasmic protein”

-          Line 482-483: the correlation between cagA positive/negative and other virulence factors is not a direct correlation calculus, but, if I intended well, it is logically consequent to the differences between sg1 and sg2. Explain better in the text

-          Line 483: oipA "on" status is not considered in Table5 and generally in all the paper. If should be mentioned in the discussion, authors must integrate.

-          Line 483: differences in HopQ types are significant in Table 2 (although the comparison should be explained better) and not significant in Table 5. Please explain/comment

-          Some typos

-          The English language of some sentences should be improved

Fig 1: contains sg3 and is not commented

Fig 2: while panels A and H contains significative information, the other panels show no difference. Since panel A is too small to be read, consider to leave the most significant panel and modify the layout. Include in the legend the statistical test employed. In Panel H, since there are differences among the groups, what condition is dissimilar? An increase of GC in this panel is not obvious to be seen

Table 2: for me it is unclear the meaning of pval. It signifies differences in the distributions of an allelic variant among the 4 clinical classes or differences within a specific clinical class? Please explain

Table 3: same issue as in Table 3. Comparisons within a specific gene are considered in rows or in column? Which test has been used?

Table 4:

-          Check/explain the set of genes for presence and completeness, they seems almost overlapping

-          Explain which comparison is considered for pval

-          ComB seems wrong

-          Why cagA is considered here only for complete cagPAI (total 172), while in Table S6 is considered in all strains? Please uniform

-          “(virB7 in instances of cagPAI” is unclear

Fig 5: some genes that are commented in the result section are missing in the figure (HP1547, HP1220)

Fig 6:

-          Enlarge Pval column

-          HP0763 row do not contained the “risk” SNP

-          It is unclear in the “Safe genotype” column the meaning of “not”: the sequence is absent? If so, how can be a synonymous mutation? Please explain

-          GluAsp135AspGly is unclear for a single mutation in position 405, please explain

-          1038_1041delGACCinsAAAT is unclear for a 1 nt SNP, please explain

-          161_163delTAAinsCAG is unclear for a 1 nt SNP, please explain

Table S1: it is odd that hypothetical proteins can be included in the list of virulence factors. Remove or add in text an appropriate explanation. Moreover, the table contains many of errors:

-          “Gene name” column contains gene name, 26695 strain name, G27 name, upper-case/lower case first letters, name in caps (SOD), some names are missing (es rpoN): please uniform and integrate

-          “Gene function” column contains gene names: remove

-          It is unclear why some genes are reported in red and with asterisks: explain in a dedicated legend

-          In the OMP section, 26695 name are wrongly reported (876 instead of HP0876)

-          Add the information of the missing HP0326 gene in the legend

Figure S1:

-          All graph can be compressed

-          a pval value is missing: add

-          text is misleading: it is not a distribution of genes (title) nor a proportion (y-axis), but, if I have intended well, the percentage of strains positive for the considered genes, with strains divided among the clinical pathologies

-          Add an information on the determination of pval for both panel A and B: it means differences in % among all the 4 conditions, or specifically for GC? What statistical analysis is used? Please explain

Table S5:

-          the final 100% is misleading, as each subsection add to 100%: please correct

Table S6 contains A LOT of errors (double check all the numbers):

-          I think comB has wrong values

-          I think cagPAI “completeness” has inverted columns

-          Tsf4b’ sum is incorrect

-          Tsf4 type: columns inverted? Correct

-          Legend is unclear: the same genes are enlisted for the “presence” and “completeness” criteria. Please explain or modify accordingly

Table S7:

-          It is unclear the “Asp235Asp” meaning if it is the table for synonymous mutations

-          “SNP type” and “Effect on amino acid sequence” columns should be removed because here they are meaningless

-          It is unclear in the “Safe genotype” column the meaning of “not”: the sequence is absent? If so, how can be a synonymous mutation? Please explain

Globally the quality of the english language is good

There are some typos (also in the title in the manuscript), and some sentences are misleading or unclear 

Reviewer 2 Report

Although H. pylori is the important carcinogen which induce the gastric cancer as notified by WHO its carcinogenicity are different depend on the strain.  so it have to be studied clearly which strains or genotypes are induce gastric cancer. 

In this points. your work is very valuable to solve the question which mentioned above.

here i leave some questions. please let me show your opinion 

1. of course the H. pylori is important to induce gastric cancer but there are many other factors like as genetic mutation of patients, environmental factor or pattern of life. when enroll the patients were you consider this points?

2. i agree the each of gene genotype is effect the carcinogenicity like as a your results. can you figure out the carcinogenicity which derived from the combination of genotypes of each  virulence factors? 

3. is there case who estimated the risk using the genotype analysis of the H. pylori?

4.  which strain is more carcinogenic between Non-GC group and GC group. is there any experimental(in vitro or in vivo) data?

Reviewer 3 Report

Saruuljavkhlan and colleagues submitted an interesting analysis of 223 H. pylori genomes from Mongolia, a high risk area for H. pylori-associated gastric cancer (GC). The genomes were sequenced from isolates obtained during esophagogastroduodenoscopy, and based on morphology and immunohistochemistry, the samples were grouped into NAG, PUD, AG or IM, and GC. Unfortunately, despite the high incidence of GC in Mongolia only 16 genomes from GC patients were included in the analysis. As the authors stated in the discussion, this was connected to the low rate of H. pylori isolation from GC samples.

The abstract and passages elsewhere in the manuscript (materials and methods, etc.) mention analysis of the secretome. However, this is misleading, because the secretome was not analysed. Instead, the authors analysed a selection of known and potential virulence factors (some of which may be secreted).

The authors analysed and described the allele composition (and presence or absence) of 12 well-known virulence factors, incl. cagA, vacA, babA, oipA, sapA, hopQ. I am a little surprised that serine protease HtrA was not included in the in depth analysis of selected virulence factors despite its importance during the H. pylori interaction with the gastric epithelium.

Using finestructure and chrom painting the Mongolian isolates were grouped into 2 groups: Sg1 (majority) and Sg2 (along with reference strain from H. pylori subpopulation hspAltai)

Not surprisingly, Sg2 was not associated with severe disease outcome , those isolates contained all the all the "harmless" virulence factor alleles (e.g. cagA-negative, vacA s2m2, hopQ type II, sabB-negative). In contrast, Sg1 was associated with GC, and those isolates contained all the highly virulent alleles (e.g. cagA-positive, vacA s1m1, hopQ type I, etc.)

Finally, the authors performed GWAS with the Sg1 group genomes and identified 9 non-syn SNPs that were associated with GC. Unfortunately, given that those SNPs were located housekeeping genes or hypothetical genes, their importance in the development of GC is unclear. In addition, no functional analyses were performed to elucidate the potential function of those SNPs.

Minor:

Title: populatio

line 164: the term HpSVF was not defined

Table 6: By definition, a non-synonymous SNP causes a single amino acid change. The SNP in HP0452 (either G or T) seems to affect amino acid tandems: GluAsp135AspGly. Please comment or correct.

discussion line 471: "We found 18 novel candidate SNPs in 16 genes by SNP-GWAS and k-mer-GWAS methods." This includes syn SNPs the relevance of which is questionable.

Line 473: Why do the authors think that a CagA prevalence of 76% is unusual? As it appears the vastly different H. pylori from Sg1 and Sg2 may belong to different phylogeographic H. pylori (sub-)populations, one of which (hspAltai) was defined based on the included reference genomes. Did the authors attempt to assign the Sg1 group to a described H. pylori sup-population? For example, Moodley et al. 2021 (doi: 10.1073/pnas.2015523118) assigned isolates from discuss the presence of ABC and ABCC EPIYA and the virtual lack of EPIYA ABD CagA in H. pylori strains from Mongolia, particularly in the light of nearly 100% EPIYA ABD CagA in the neighboring country China.

The language needs editing, the meaning of several sentences is unclear.

For example line 477: "Especially, we found that, in contrast to other Western countries, all Sg1 strains in Mongolia exhibit Western-type CagA with complete cagPAI, much as in East Asian countries." Please split this sentence, as the cagPAI in H. pylori from East Asian countries does not contain the Western-type CagA.

Round 2

Reviewer 1 Report

Almost all the points have been addressed.

I suggest only a minor revision of Table S1, as

- "gene function" column still contains all the gene names as in "gene name" column (point 33 was actually not addressed)

- it is unclear why some functional groups are separated into diffrent subgroups (e.g. "flagellar component" is from rows 12 to 62, and again from 199 to 203 and from 204 to 209)
